# Systematic Investigation of Structural, Morphological, Thermal, Optoelectronic, and Magnetic Properties of High-Purity Hematite/Magnetite Nanoparticles for Optoelectronics

**DOI:** 10.3390/nano12101635

**Published:** 2022-05-11

**Authors:** Akbar Ali Qureshi, Sofia Javed, Hafiz Muhammad Asif Javed, Muhammad Jamshaid, Usman Ali, Muhammad Aftab Akram

**Affiliations:** 1School of Chemical & Materials Engineering, National University of Sciences & Technology, Islamabad 44000, Pakistan; aali.phdscme@student.nust.edu.pk (A.A.Q.); usman.phdscme@student.nust.edu.pk (U.A.); aftabakram@scme.nust.edu.pk (M.A.A.); 2Department of Mechanical Engineering, Bahauddin Zakariya University, Multan 60000, Pakistan; muhammad.jamshaid@bzu.edu.pk; 3Department of Physics, University of Agriculture Faisalabad, Faisalabad 38000, Pakistan; m.asif.javed@uaf.edu.pk

**Keywords:** hematite, magnetite, top-down, optoelectronics

## Abstract

Iron oxide nanoparticles, especially hematite (α-Fe_2_O_3_) and magnetite (Fe_3_O_4_) have attained substantial research interest in various applications of green and sustainable energy harnessing owing to their exceptional opto-magneto-electrical characteristics and non-toxicity. In this study, we synthesized high-purity hematite and magnetite nanoparticles from a facile top-down approach by employing a high-energy ball mill followed by ultrasonication. A systematic investigation was then carried out to explore the structural, morphological, thermal, optoelectrical, and magnetic properties of the synthesized samples. The experimental results from scanning electron microscopy and X-ray diffraction corroborated the formation of highly crystalline hematite and magnetite nanoparticles with average sizes of ~80 nm and ~50 nm, respectively. Thermogravimetric analysis revealed remarkable results on the thermal stability of the newly synthesized samples. The optical studies confirmed the formation of a single-phase compound with the bandgaps dependent on the size of the nanoparticles. The electrochemical studies that utilized cyclic voltammetry and electrochemical impedance spectroscopy techniques verified these iron oxide nanoparticles as electroactive species which can enhance the charge transfer process with high mobility. The hysteresis curves of the samples revealed the paramagnetic behavior of the samples with high values of coercivity. Thus, these optimized materials can be recommended for use in future optoelectronic devices and can prove to be potential candidates in the advanced research of new optoelectronic materials for improved energy devices.

## 1. Introduction

The exceptional opto-electrical characteristics of metal oxide nanostructures relative to their bulk counterparts have gained substantial attention in various applications of energy conversion/storage devices such as solar cells, supercapacitors, LEDs, Li-ion batteries, electrochemical sensors as well as hydrogen production [1,2,3,4,5,6,7]. Among others, iron-oxide-based nanostructures have proven to be potential candidates owing to their outstanding thermal stability, high surface area, chemical stability, non-toxicity, and superior opto-magneto-electrical properties [8,9,10,11]. Iron oxides are naturally occurring mineral compounds existing in three well-known polymorphic forms, including hematite (α-Fe_2_O_3_), maghemite (γ-Fe_2_O_3_), and magnetite (Fe_3_O_4_) [12]. The hematite (α-Fe_2_O_3_) phase of iron oxide is thermodynamically stable with a narrow bandgap of (~2–2.3) eV, easier to synthesize from natural resources, and stable in aqueous solution [13,14,15,16], while on the other hand, the magnetite (Fe_3_O_4_) phase differs due to its unique characteristics, mainly of superparamagnetic behavior, high coercivity, biocompatibility, mixed-valence of bivalent (Fe^2+^) and trivalent (Fe^3+^) ions, and perfect conductivity [17,18,19,20]. The superior opto-magneto-electrical properties of hematite and magnetite nanostructures have attracted the scientific community to explore their features for mass production to meet energy-related demands [21]. Hematite and magnetite nanoparticles have been synthesized using various conventional synthesis routes reported in the literature mainly involving co-precipitation, sol–gel, sonolysis, polyol, hydrothermal, and microwave-assisted synthesis [22,23,24,25,26,27]. Amongst the above bottom-up approaches, co-precipitation is considered to be a widely adopted technique by researchers. However, the difficulty in controlling oxidation and maintaining pH are always the limiting factors present during the synthesis. Another drawback of using these approaches is that to synthesize even a single gram of nanopowder, a lot of resources, human effort, and time is required, resulting in a limited and expensive product that simply cannot fulfill the present demand. Moreover, many of these chemical methods involve toxic precursors or are based on complex solution processes. The use of hazardous chemical regents during synthesis may also limit their application involving desalination or wastewater treatment plants [28]. To overcome these critical challenges, a low-cost and non-toxic approach is desirable for the mass production of iron oxide nanoparticles. Instead of the bottom-up techniques, the top-down approach presents a facile, eco-friendly synthetic route including crushing, milling, and grinding [29,30]. From the practical point of view, ball milling is a facile, fast, cost-effectual green technology with massive potential. A high-energy ball mill is utilized to produce nanopowders from micron- or sub-micron-size powders [31]. Moreover, the milling parameters involving speed and time have a significant impact on the morphology and crystallinity of synthesized powders. Limited efforts have been devoted to the resolution of the issues of cost, yield, quality, and performance so far. The facile top-down approach is also beneficial in the mega-scale production of nanopowders from naturally occurring mineral deposits. Our region, Pakistan has plenty of reserves of naturally occurring minerals, especially hematite and magnetite, but is poor in development [32]. Therefore, it is the need of the hour to explore the potential of these natural minerals for their utilization in optoelectronic devices. In this way, low-cost future optoelectronic devices can be fabricated by utilizing natural resources for the fulfillment of the current global energy demand. Surprisingly and interestingly, such a low-cost, nontoxic synthesis approach by utilizing natural ores of iron oxide for the synthesis of high-quality hematite and magnetite nanoparticles with reasonable time and resources has not been previously reported.

Here, we report the synthesis of hematite and magnetite nanoparticles from naturally occurring, high-purity, micron-size ore powders by using a top-down approach involving ball milling of powder followed by ultrasonication [33]. The current synthesis procedure suggests that this method is facile and cost-efficient because it entails minimum materials and energy to produce nanoparticles with similar characteristics to those synthesized from bottom-up approaches. Furthermore, a systematic study was conducted to examine the morphology, crystallinity, and thermal stability of nanoparticles. Moreover, opto-magneto-electrical properties were also analyzed to prove these iron oxide nanoparticles as prospective candidates for future optoelectronic devices. The presented facile, novel, and industrially sustainable techniques are judiciously required for the fulfillment of the current demand for iron oxide nanoparticles. Despite intensive studies on the synthesis and characterization of iron oxide nanoparticles, there is no combined report available regarding the systematic investigation of the structural, morphological, thermal, and opto-magneto-electrical properties of high-purity hematite and magnetite nanoparticles to the best of the author’s knowledge.

## 2. Materials and Methods

### 2.1. Materials

The natural high-purity ores of iron oxides, hematite (α-Fe_2_O_3_), and magnetite (Fe_3_O_4_), in micro-meter-size powder form, were acquired from the Pakistan Council of scientific and industrial research (PCSIR), Pakistan. Ethanol, 2-propanol, and distilled water were used for all the suspension preparations. The samples were utilized as received with no chemical modifications unless stated otherwise.

### 2.2. Synthesis of Hematite (α-Fe_2_O_3_) and Magnetite (Fe_3_O_4_) Nanoparticles

For the synthesis of hematite (α-Fe_2_O_3_) and magnetite (Fe_3_O_4_) nanoparticles, the conventional top-down approach by using a high-energy ball mill (Fritsch Pulverisette 6) was chosen. The micro-meter-size powder of iron oxide samples was put into a vial. The 28 stainless steel balls of 9 mm diameter were utilized to grind the samples. The ball mill was operated at a speed of 300 rpm for 6 h with the ball to powder ratio (BPR) of 10:1. After every half an hour, the ball mill was paused for 15 min to avoid overheating. After 6 h, the sample was taken out, and a small quantity of the ball-milled powder was mixed with 2-propanol. The mixture of 2-propanol and powder sample (~1% by mass) was placed under an ultrasonic homogenizer (JY92-IIN) operating at a power of 400 W for 75 min at room temperature. After eight hours of break, ~35% of the mass was accumulated suspended as supernatant, and dried under a vacuum oven at 50 °C. The dried sample was then used further for the required characterizations. Appendix A depicts the ball-milled samples of magnetite (Fe_3_O_4_) and hematite (α-Fe_2_O_3_) nanopowders.

### 2.3. Measurement and Characterization

The elemental compositions of the natural hematite (α-Fe_2_O_3_) and magnetite (Fe_3_O_4_) micro-meter-size powders were examined using an X-ray Fluorescence Spectrometer (Philips PW 2400, Labexchange, Burladingen, Germany). The surface morphology of iron oxide powders was investigated before and after ball milling using a VEGA3 (Tescan, Warrendale, PA, USA) scanning electron microscope equipped with an EDS detector. To identify the size distribution of the sample powders, a laser particle size analyzer (HORIBA LA-920, Horiba Scientific, Kyoto, Japan) was utilized. The crystalline structure of the iron oxide powders was analyzed using a D8 Advance XRD diffractometer (Bruker, Karlsruhe, Germany) in the 2θ range of 10°–70°. The Raman spectra were recorded by using a Raman spectrophotometer (BWS415-532S, Plainsboro, NJ, USA). To investigate the thermal stability of hematite and magnetite powders, the samples were scanned up to 800 °C by using a DTG-60 analyzer (Shimadzu, Kyoto, Japan) at a ramp rate (10 °C/min). The UV–Vis absorption spectra were recorded to investigate optical properties by using a spectroscope (JENWAY, Stone, UK), and the bandgap was measured by drawing tauc plots. The PL spectrofluorometer (Fluoromax-4, Horiba Scientific, Kyoto, Japan) was deployed to measure photoluminescence emission at an excitation wavelength of 385 nm. The electrochemical measurements (CV, EIS, and Tafel) were performed on a two-electrode electrochemical workstation (Bio-Logic VSP-1086, Seyssinet-Pariset, France) utilizing a 1 M KOH electrolyte. The hall effect of the samples was measured to examine the charge carrier’s mobility, conductivity, and bulk concentration on a four-probe sample holder by using an electromagnet. The measurements were executed several times, and the average values were reported. Finally, the magnetic properties of hematite and magnetite samples were studied under ambient conditions by using a vibrating sample magnetometer (Lake Shore Cryotronics 7400 Series VSM, Lake Shore Cryotronics, Westerville, OH, USA).

## 3. Results

### 3.1. Elemental and Structural Properties of Hematite (α-Fe_2_O_3_) and Magnetite (Fe_3_O_4_) Powders

The elemental composition of hematite (α-Fe_2_O_3_) and magnetite (Fe_3_O_4_) micro-powders was studied using X-ray fluorescence analysis. Appendix A represent the percentage of different elements present in hematite and magnetite samples. As evident from XRF analysis, the ore micro-powders were of high grade with plentiful iron contents. It is also evident from the results that the XRF technique could not distinguish the presence of Fe^2+^ and Fe^3+^ ions in the samples; rather, it would only ascertain the net amount of iron in each sample [34]. Moreover, XRF analysis revealed a high content of iron (above 95% by mass) with a very small mass percentage of other impurities for both samples, indicating high purity.

The structure, phase identification, crystallinity, and crystallite size of ball-milled hematite and magnetite nanopowders were verified using the XRD technique as depicted in Figure 1a,b. The 2θ peaks at 24.13°, 33.14°, 35.60°, 40.85°, 49.43°, 54.04°, 57.57°, 62.41°, and 63.98° in Figure 1a were assigned to (012), (104), (110), (113), (024), (116), (122), (214), and (300) planes of rhombohedral hematite belonging to space group R-3c (JCPDS card no. 89-0596). Similarly, the prominent 2θ peaks at 30.08°, 35.43°, 43.09°, 56.99°, and 62.60° in Figure 1b were assigned to (220), (311), (400), (511), and (440) planes of inverse cubic spinel magnetite belonging to space group Fd-3m (JCPDS card no. 89-3854). The strong and sharp diffraction peaks indicate the high crystallinity and homogeneity of these samples. No diffraction peaks other than hematite and magnetite were observed in XRD diffractograms, indicating the phase purity of the samples. The average crystallite sizes of the hematite and magnetite nanopowders were evaluated by using the Debye–Scherrer equation ($D=0.9\lambda /\left(\beta cos\theta \right))$ [35]. The calculation for the crystallite sizes was executed for the dominating peaks of highest intensity positioned at 33.13° consistent to the plane (104) for hematite and 35.43° consistent to the plane (311) for magnetite, respectively. Then, by using XRD data analysis, the interplanar spacing d *_hkl_*, average crystallite size, and cell volume for both the samples were also evaluated and are listed in Appendix A. The average crystallite size for the most intense peaks of hematite and magnetite was found at around 92 nm and 44 nm, respectively. The results show that the facile top-down approach by use of ball mill followed by ultrasonication could produce nanometric-size powder consisting of highly crystallized α-Fe_2_O_3_ and Fe_3_O_4_ particles, respectively.

To further investigate the structural aspects of hematite and magnetite nanoparticles, Raman spectroscopic measurements were performed. Figure 1c,d depicts the Raman spectra of hematite and magnetite nanoparticles. In hematite, ions are closely packed in a hexagonal structure with arrays of O atoms, while Fe has an octahedral structure. As O^2−^ anions have larger ionic radii as compared to Fe^3+^ cations, they prefer hexagonal sites. It has been well understood and reported that magnetite light scattering is less than hematite and computationally simulated Raman spectra of hematite and magnetite show peaks at 790 cm^−1^. It has been widely reported that pure Fe_3_O_4_ particles have size-dependent behavior, which can be attributed to the oxidation-induced lattice strain. Moreover, the intermediate maghemite peak is present at 918 cm^−1^. It has also been reported that the characteristic Raman peak of Fe-O-Fe occurs at around 900 cm^−1^ [36]. Such oxo-bridge di-iron bonds happen at the interface of magnetite and the maghemite phases of iron oxide as both have a spinel crystal structure. For hematite, peaks are present at 145 cm^−1^_,_ 455 cm^−1^_,_ 790 cm^−1^, and 1090 cm^−1^, while for magnetite, peaks are present at 318 cm^−1^ and 918 cm^−1^. These peaks demonstrated a strong response against incident light and have been widely supporting their use due to their optoelectronic properties [37].

### 3.2. Surface Morphological Characteristics of Hematite (α-Fe_2_O_3_) and Magnetite (Fe_3_O_4_) Nanoparticles

Scanning electron microscopy (SEM) analysis was performed to examine the morphology and particle sizes of hematite and magnetite powders before and after ball milling. Appendix A depict the SEM micrographs of hematite and magnetite micro-size powders before ball milling up to different resolution ranges, respectively. The laser particle size analyzer was deployed initially to examine the sizes of hematite and magnetite powders and the results are displayed as histograms in Appendix A. Similarly, Figure 2a–d, and Figure 3a–d manifest the SEM micrographs of hematite and magnetite powders after ball mill. Furthermore, energy-dispersive spectroscopy (EDS) analysis was conducted to confirm the atomic and weight percentages of iron and oxygen in hematite and magnetite ball-mill-treated samples, and the results are displayed in Appendix A.

Initially, the procured powders of hematite and magnetite were in the micrometer range, as verified with the particle size analyzer. To further investigate the morphology of pure hematite powder, the particles in Appendix A are found in micron sizes with agglomeration and possess random morphology including spherical, non-spherical, and tabular particles. Similarly, in Appendix A, the particle size for magnetite micro-powder is in the micrometer range, as indicated by the particle size analyzer. Moreover, these particles are also of random shapes. Therefore, to investigate their exceptional characteristics in optoelectronics, size reduction is crucial [38]. At the nanoscale, various opto-magneto-electrical properties change due to a reduction in the size of the particles. Therefore, this micron-size powder was subjected to a ball mill followed by ultrasonication to attain a reduction in the sizes of these bigger particles.

Figure 2a–d depicts the morphology of hematite nanoparticles after the ball milling process. The particles are observed to be seriously agglomerated due to the absence of any surfactant used to overcome the agglomeration issues. Therefore, the specific shape and accurate size of the individual nanoparticle is difficult to be determined accurately due to lump formations and agglomerations. Moreover, these clusters result in increased particle sizes. However, the particles of the hematite sample are observed to be in the nanometric range below 100 nm. The high surface energy of iron oxide nanoparticles owing to their large specific surface area tends to usually agglomerate in the form of clusters. Figure 3a–d depicts the morphology of magnetite ball mill powders. The particles are homogeneously distributed and spherical. It is established that the growth of particles is governed by their isotropic unit cell structure leading to spherical or non-spherical shapes. The size variation, size distribution, and the shape of the particle can significantly alter the properties of nanomaterials. Similarly, the optoelectrical characteristics involving charge transfer, concentration, mobility, and conductivity strongly depend upon these factors. A slight variation in size and shape may cause significant enhancement in the above-mentioned factors [39,40]. Magnetite nanoparticles have an inverse spinel cubic structure, and they have a strong tendency to form aggregates due to high surface energy and strong magnetic dipole–dipole interaction [41]. The average size of nanoparticles is ~50 nm. Slightly less agglomeration with spherical morphology was observed for magnetite nanoparticles relative to hematite nanoparticles. The particle size measured using SEM analysis for hematite and magnetite is seen to be different compared to the crystallite size measured using XRD. This difference in size may be due to the fact that a particle size may consist of more than one crystallite.

Appendix A depict the EDS spectrum of ball-milled hematite nanoparticles, while Appendix A depict that of magnetite nanoparticles. The EDS spectrum only consists of peaks of Fe and O for both samples. No impurity peak was detected for both samples, which is consistent with the XRD results.

### 3.3. Thermal Stability of Hematite (α-Fe_2_O_3_) and Magnetite (Fe_3_O_4_) Nanoparticles

The thermal stability of hematite and magnetite nanopowders was investigated in the temperature range from 28 to 800 °C using a thermo-gravimetric analyzer (TGA).

Figure 4a,b depicts the weight percentage change as a function of the temperature for both samples. Moreover, samples were subjected to a maximum temperature of 800 °C at a ramp rate of 10 °C/min under an inert atmosphere to examine degradation. The mass of the samples was ~21.5 mg. The TGA plot for hematite nanoparticles suggests two decomposition steps. The preliminary weight loss of 7% up to 300 °C was possibly due to the exclusion of surface-adsorbed water, while the weight loss at higher temperatures (300 °C—onwards) could be mainly ascribed to evaporation, the partial dehydration of structural water, or due to the presence of some impurities [42,43]. The abrupt drop in weight above 500 °C might be due to the decomposition of Fe_2_O_3_ into Fe_3_O_4_. The reduction in hematite to magnetite at elevated temperatures might be the reason behind more weight loss as compared to the magnetite sample [44]. The magnetite nanoparticles in Figure 4b demonstrated excellent thermal stability up to 800 °C, and only minor weight loss of ~1% was observed, which could be due to the moisture effects of adsorbed water lost steadily throughout the whole process. This indicates the high purity of magnetite nanopowder. Overall, both samples of high-purity natural iron oxides, hematite and magnetite, demonstrated outstanding thermal stability over a wide range of temperatures and were proven to be thermally stable candidates for optoelectronic applications.

### 3.4. Absorption Characteristics of Hematite (α-Fe_2_O_3_) and Magnetite (Fe_3_O_4_) Nanoparticles

UV–Vis spectroscopic measurements were performed to examine the optical properties and electronic structure of iron oxide nanoparticles, which are crucial aspects to contemplate in selecting appropriate materials for optoelectronic devices. Figure 5a,b depict the UV–Vis absorption spectra of hematite and magnetite nanoparticles, respectively. The optical bandgap energies of hematite and magnetite nanopowders were measured by plotting (*αhν*)^2^ vs. (*h*ν) photon energy using Tauc’s equation: ${\left(\alpha h\nu \right)}^{n}=A\;\left(h\nu -E_{g}\right)$, and the corresponding values of energy bandgaps E*_g_* were determined and manifested as insets of Figure 5a,b for hematite and magnetite, respectively. The single absorption peak was observed for both samples of iron oxides, verifying the formation of a single-phase compound.

Some reports suggested that α-Fe_2_O_3_ shows shreds of evidence of both direct as well as indirect bandgap material. The spin forbidden Fe^3+^ 3d → 3d excitation causes indirect transition, while O^2−^ 2p → Fe^3+^ 3d charge transfer leads to direct transition. Moreover, the α-Fe_2_O_3_ nanoparticles have a bandgap of 2.65 eV as compared to their bulk counterpart (2.2 eV). In our results, the bandgap of hematite nanoparticles was found at around 2.66 eV, confirming the successful formation of hematite nanoparticles. The UV–Vis absorption spectrum of magnetite exhibits an absorption band in the region of 340–430 nm. The bandgap of magnetite nanoparticles was found at around 2.2 eV, which agrees with previous studies [45]. The energy bandgaps of nanostructures are dependent on the size of the nanoparticles, and hence, by controlling sizes, different bandgaps are expected [18]. The values of bandgaps for both the samples confirmed them to be semiconductor materials due to their bandgaps lying in the range of semiconductor energy bandgaps (0–3 eV) [46].

The steady-state photoluminescence (PL) emission measurements were conducted at an excitation wavelength of 385 nm, and the results are manifested as (c) for hematite and magnetite nanoparticles. As reported in the literature, bulk hematite does not show the photoluminescence response [47,48] as it has d-d forbidden transition, magnetic relaxation, and resonant charge transfer between cations. However, the nanostructures of hematite show photoluminescence because of self-trapped states which are associated with their optical properties. In the nano-regime, quantum confinements play a vital role in the delocalization and quantization of electronic states, which ultimately relaxes the forbidden rule of d-d transition. Further, the reduced particle size minimizes the magnetic interactions between the cations as the loss of long-range ferromagnetic order. Thus, partially optical transition is allowed. As shown in Figure 5c, strong PL was observed from hematite nanoparticles dispersed in water and coated as the film on the glass slide. These nanoparticles show strong PL emissions at 571 nm. This dominant peak can be attributed to the band edge emission. The peak emission at 571 nm is attributed to ^6^A_1g_ → ^4^T_1g_ and d–d transition of Fe^3+^ ions owing to the crystal field splitting of a FeO_6_ octahedron with O_h_ symmetry [49]. Furthermore, for the magnetite nanoparticles, peak intensity is decreased and blue-shifted for Fe_3_O_4_. Mostly, nanoparticles’ defects reduce the PL intensity and broaden the emission peaks. Additionally, the surface defects alter the electronic structure, leading to the narrowing of the bandgap. In this case, the non-radiative movement of electrons between different defect sites is caused by the release of thermal energy. In the present study, the transition of trapped electrons in different defect sites results in peaks around 690 nm and 750 nm.

### 3.5. Electrochemical Measurements of Hematite (α-Fe_2_O_3_) and Magnetite (Fe_3_O_4_) Nanoparticles

The cyclic voltammograms for hematite (α-Fe_2_O_3_/GCE) and magnetite (Fe_3_O_4_/GCE) nanoparticles modified glassy carbon electrode (GCE) versus Ag/AgCl as the reference electrode were recorded at a scan rate of 50 mV/s by using 1 M KOH electrolyte between −1.4 and +0.6 V and are displayed in Figure 6a,b. The overall CV patterns of both the iron oxide samples consist of a pair of anodic and cathodic peaks representing the oxidation and reduction reactions. The narrow peak-to-peak splitting (Epp) indicated the better electrochemical activity of the iron oxide samples necessary for redox reactions. In the cathodic scan, two well-defined peaks were observed due to the electrochemical reduction of Fe^3+^ and Fe^2+^ to Fe^0^, while in the anodic scan, again, two broad peaks were observed, attributed to the oxidation of Fe^0^ to Fe^2+^ and Fe^3+^, respectively [50]. The α-Fe_2_O_3_/Fe_3_O_4_ nanoparticles modified GCE demonstrated a better current response due to the presence of electroactive species on the electrode surface. Moreover, α-Fe_2_O_3_/Fe_3_O_4_ nanoparticles can act as a charge transfer medium and enhance the charge transfer process during an electrochemical reaction [51,52]. The introduction of α-Fe_2_O_3_/Fe_3_O_4_ nanoparticles on the GCE surface facilitated the conduction pathway by exhibiting reversible behavior and possessed faster charge transfer kinetics as indicated by low Δ𝐸𝑝 values (0.16 V for α-Fe_2_O_3_ and 0.21 for Fe_3_O_4_). However, further studies are required to examine the redox process kinetics in detail, as our primary goal in this study is to corroborate the potential of iron oxides (hematite/magnetite) in optoelectronics.

Figure 6c,d represent the EIS spectra of hematite and magnetite nanoparticles. It has been widely known that EIS is an interesting technique for the analysis of the interaction process of different molecular structures. The physical, chemical, and mechanical properties of the material are due to the intrinsic organization of its constituents with dielectric relaxation, which affects the dipole moments present in the material [53]. It can reveal useful information about the characteristic molecular arrangements. For the better analysis of the microscopic organization of materials, EIS studies were carried out for the present samples. As we anticipated, the impedances for magnetite are high: at the low-frequency range, the semicircle is due to grain boundaries at the surface morphology, and the predominant capacitive behavior of materials is evident from the results [54]. This predominant capacitive behavior can be observed throughout the frequency range. At higher frequencies, the resistance of the solution also contributes towards the impedance, and at the lowest ones, the charge transfer resistance is divulged [55]. The charge transfer resistance is assigned to the neglected reaction of oxide groups present at the surface. The equivalent circuit fitting is shown in the figure and is fitted by using a Levenberg–Marquardt algorithm. In this circuit, the electrolyte resistance (R1) is in series with charge transfer resistance (R2), and in parallel with the constant phase element (CPE1), which in this case represents the non-ideal double-layer capacitance.

Tafel polarization studies were carried out to observe the electrocatalytic response of iron oxide nanoparticles to measure optoelectronic as well as electrical properties. As shown in the Figure 6e,f, the plot consists of three main regions: the diffusion region at high potential, the polarization region at low potential, and the region in between them is named the Tafel region. The J_o_ value for magnetite is −1.7 mA.cm^−3^ which is inversely proportional to charge transfer resistance; from the impedance spectroscopy, it is evident that magnetite has higher charge transfer resistance as compared to hematite, which has low charge transfer resistance [56]. The values of J_o_ for hematite are low, and these results are completely in the agreement with EIS results. The electrical properties of hematite (α-Fe_2_O_3_) and magnetite (Fe_3_O_4_) nanoparticles were investigated using hall effect measurements by spin coating the α-Fe_2_O_3_/Fe_3_O_4_ thin film on a non-conductive glass substrate, and the corresponding values are listed in Appendix A. The experimental values of conductivity and mobility are in close resemblance with previous findings. Both the samples demonstrated superior charge carrier concentrations with reasonable conductivity and the mobility necessary to transport the charge carriers. The measurements were repeated multiple times, and the average values are reported. Thus, from the results, hematite and magnetite nanoparticles can be considered future potential low-cost materials in optoelectronics with significant characteristics.

### 3.6. Magnetic Properties of Hematite (α-Fe_2_O_3_) and Magnetite (Fe_3_O_4_) Nanoparticles

The magnetic properties of hematite and magnetite nanoparticles were investigated by using VSM at a temperature of 300 K under an external magnetic field ranging between −10 kOe and +10 kOe. Figure 7a,b depicts the M–H hysteresis loop for the two samples, and the corresponding values of saturation magnetization Ms (emu/g), coercivity Hc (Oe), and the remanent magnetization Mr (emu/g) are listed in Appendix A. The hysteresis loop reveals that the hematite nanoparticles have paramagnetic behavior with an Ms value of 1.242 emu/g, while the magnetite nanoparticles have paramagnetic (approximately near superparamagnetic) behavior with an Ms value of 70.34 emu/g. The saturation magnetization values for both the samples are, however, somewhat lower than that of bulk materials, which may be ascribed to the presence of lattice defects caused by the grinding process during the ball milling of the samples [14,25]. The increased coercivity values for hematite and magnetite nanoparticle samples can be partially ascribed to a surfactant-free surface since no surfactant was utilized during this research work, or they may be due to the presence of nanoparticles aggregates leading to high values of coercivity. Furthermore, the development of a saturated loop corroborated the magnetic nature of both samples [57]. Usually, the magnetic characteristics of materials depend upon various factors such as morphology, particle size, and crystal structure. The slight variation in the particle sizes can induce unusual magnetic behavior which would be quite different to those of bulk materials, which probably precede a wide range of magnetic properties between superparamagnetism, paramagnetism, ferromagnetism, and ferrimagnetism [58].

## 4. Discussion

Iron oxide nanoparticles have been widely used owing to their facile synthesis, controllable sizes, and non-toxic nature and have been considered as potential candidates for green energy applications. The top-down and bottom-up approaches are recognized approaches for the synthesis of high-quality and high-purity iron oxide nanoparticles. Yet, the yield with a bottom-up technique is low. The low-cost synthesis of iron oxide nanoparticles with the top-down approach is essential to meet the demand for the mega-scale production of optoelectronic devices. We synthesized iron oxide, hematite, and magnetite nanoparticles with natural ore powders using a facile ball mill approach followed by ultrasonication. The elemental compositions in Appendix A reveal that ore powders are found to be plentiful in iron contents with high purity. We also noted from SEM images (Appendix A) that the size of the natural iron oxide particles is large before being subjected to a ball mill process. The size reduction is crucial for the application of these natural ore powders in optoelectronics. The simple ball mill technique significantly reduced the sizes of natural ore powders up to the nanometer scale (Figure 2 and Figure 3). The particles were observed to have random morphologies with spherical and non-spherical shapes. The strong, sharp diffraction peaks of hematite and magnetite nanoparticles in Figure 1a,b with no impurities reflect the high crystallinity of these nanoparticles. From XRD analysis, we also confirmed the rhombohedral and inverse cubic spinel structures for hematite and magnetite, respectively. Furthermore, Raman spectra (Figure 1c,d) also confirmed the purity and types of phases present, following the literature for hematite and magnetite. We also noted that weak Raman scattering for magnetite nanoparticles as compared to hematite may be due to laser-induced phase transformation. Since the synthesis of nanoparticles was conducted without utilizing any surfactant, we also observed agglomeration and cluster formation. The degradation of materials upon exposure to heat and temperature is also a limiting factor for their application in optoelectronics. We noted the formation of very stable, non-degradable iron oxide nanoparticles with minor weight loss up to higher temperatures (Figure 4). We observed that employing a simple top-down approach and the utilization of natural ore powders can produce low-cost nanoparticles with high yield relative to bottom-up techniques. After ensuing substantial structural, morphological, and thermal properties, the optical studies were investigated as the foremost step for establishing the application of nanoparticles in optoelectronics.

The reduction of particle sizes may result in bandgap variation, which we noted (Figure 5a,b); the bandgaps calculated using tauc plots were found to be different as compared to bulk counterparts. Eventually, by controlling the sizes, the bandgap can be varied, which is indeed an important aspect in applications involving thin-film devices such as solar cells, Li-ion batteries, or many other applications. The absorption decreases in the visible region for both hematite and magnetite, indicating that these materials can be used as charge transport materials with any absorber. UV–Vis spectra also exhibited the formation of single-phase compounds previously confirmed using XRD and Raman analysis. The PL emission spectrum of the nanoparticles helped in gaining insights into structure-related defects and electron–hole recombinations. The hematite PL spectra have a strong emission peak due to band edge emission, while magnetite exhibits somewhat lower intensity in PL (Figure 5c). Surface defects alter the electronic structure, resulting in narrowing bandgaps and wide emission peaks. Higher PL intensity values confirm the metallic nature of these high-purity nanoparticles. We also discussed the electrocatalytic properties of these nanoparticles. The nanoparticles are electroactive with reversible redox natures with smaller Epp values (Figure 6a,b). The superior charge carrier concentration was observed for the iron oxide nanoparticles from hall effect measurements (Appendix A) with low charge transfer resistances (Figure 6c,d). The iron oxides at bulk have usually low mobilities but the size reduction of these high-purity nanoparticles has also resulted in improved conductivities and mobilities vital in optoelectronics applications. Magnetic properties were observed for in nanoparticle samples with a high value of coercivity (Figure 7) since magnetic properties are not often studied in optoelectronic devices, yet these high-purity nanoparticles with promising values of saturation magnetization are recommended for other applications in ferrofluid and biomedical fields. Further, particle sizes also influence the magnetic properties, and a decrease in size may result in unusual magnetic characteristics different from the bulk. Usually, Ms decreases with decreasing crystallite size for mono-domain particles, yet for our samples, we did not notice any significant difference.

## 5. Conclusions

In summary, we synthesized high-purity hematite (α-Fe_2_O_3_) and magnetite (Fe_3_O_4_) nanoparticles with a facile top-down approach using a high-energy ball mill followed by ultrasonication. The experimental results revealed the formation of highly crystalline and non-degradable hematite and magnetite nanoparticles with no peaks of impurity. The SEM images verified the formation of nanoparticles with spherical and non-spherical morphologies. The optical studies confirmed the formation of a single-phase compound with bandgaps of 2.66 eV and 2.2 eV for hematite and magnetite nanoparticles, respectively, indicating the bandgap dependence on the size of the nanoparticles. The strong PL emission peak for hematite and magnetite suggested fewer defects and trap states. The electrochemical studies confirmed these iron oxide nanoparticles as electroactive species supporting the charge transfer process. The charge concentration, mobility, and conductivity of high-purity iron oxide nanoparticles were remarkably superior to their bulk counterparts, confirmed using hall effect measurements. The magnetic studies revealed the paramagnetic behavior of the samples with the saturated magnetization of 1.24 emu/g and 70.34 emu/g and coercivity of 246 Oe and 149.3 Oe for hematite and magnetite nanoparticles, respectively. The results of this study suggested that these low-cost and highly pure hematite (α-Fe_2_O_3_) and magnetite (Fe_3_O_4_) nanoparticles could be promising candidates for the fabrication of future optoelectronic devices.

## Figures and Tables

**Figure 1 nanomaterials-12-01635-f001:**
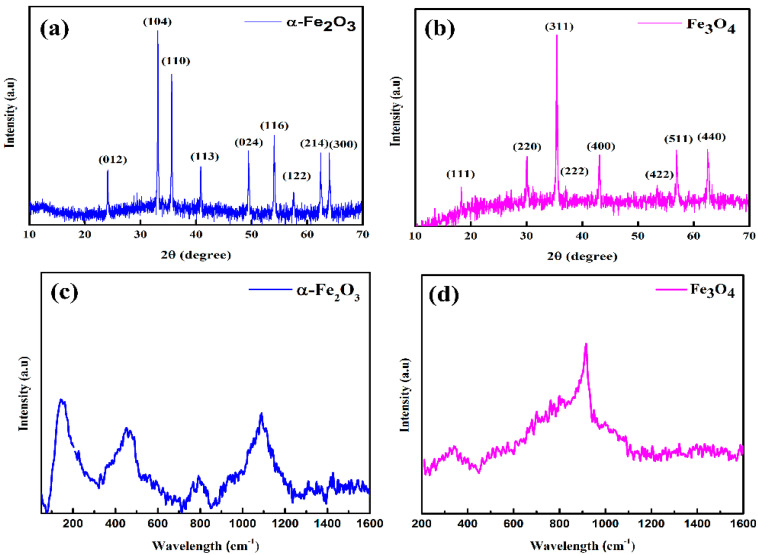
XRD diffractograms of (**a**) hematite (α-Fe_2_O_3_) and (**b**) magnetite (Fe_3_O_4_) nanopowders; Raman spectra of (**c**) hematite (α-Fe_2_O_3_) and (**d**) magnetite (Fe_3_O_4_) nanoparticles.

**Figure 2 nanomaterials-12-01635-f002:**
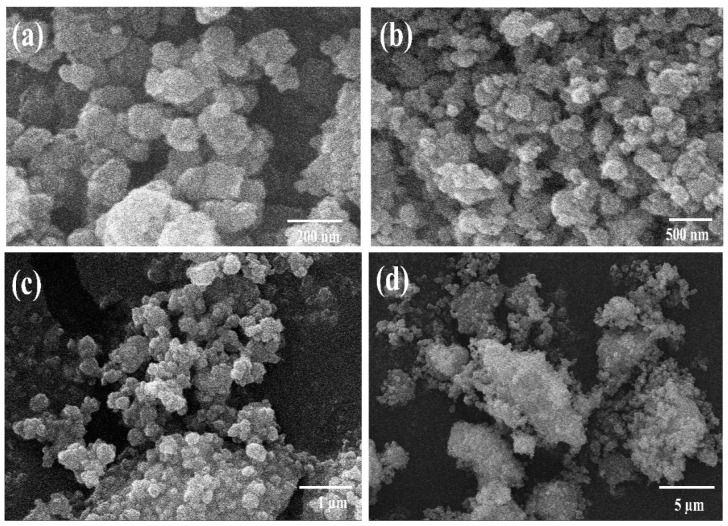
SEM images of the hematite (α-Fe_2_O_3_) nanoparticles with varying magnification, (**a**) 200 nm, (**b**) 500 nm, (**c**) 1 µm and (**d**) 5 µm after ball milling.

**Figure 3 nanomaterials-12-01635-f003:**
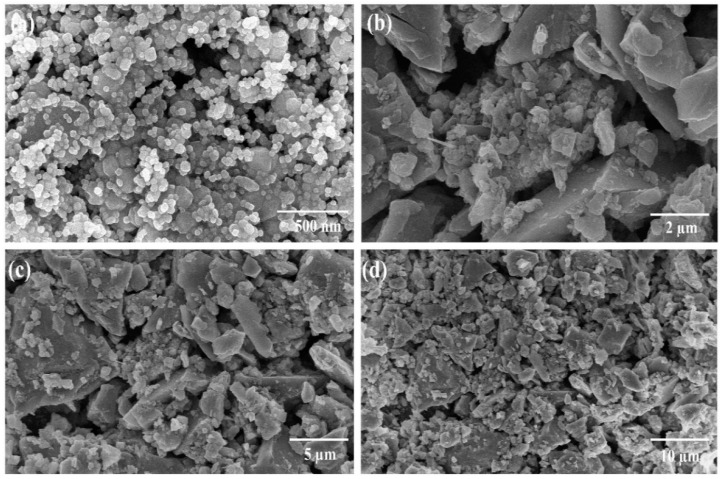
SEM images of the magnetite (Fe_3_O_4_) nanoparticles with varying magnification, (**a**) 500 nm, (**b**) 2 µm, (**c**) 5 µm and (**d**) 10 µm after ball milling.

**Figure 4 nanomaterials-12-01635-f004:**
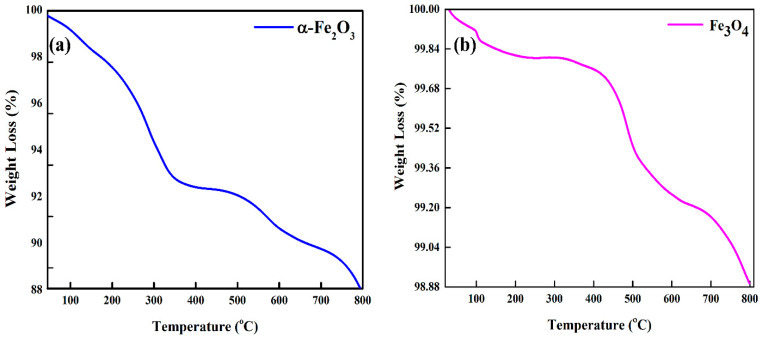
TGA curve of (**a**) hematite (α-Fe_2_O_3_) and (**b**) magnetite (Fe_3_O_4_) nanoparticles.

**Figure 5 nanomaterials-12-01635-f005:**
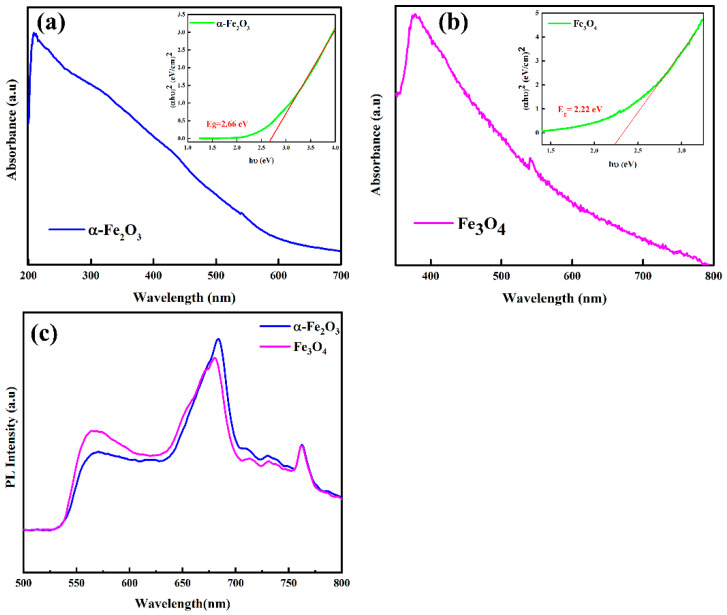
UV–Vis absorption spectra of (**a**) hematite (α-Fe_2_O_3_) and (**b**) magnetite (Fe_3_O_4_) nanoparticles; PL spectra (**c**) of hematite (α-Fe_2_O_3_) and magnetite (Fe_3_O_4_) nanoparticles.

**Figure 6 nanomaterials-12-01635-f006:**
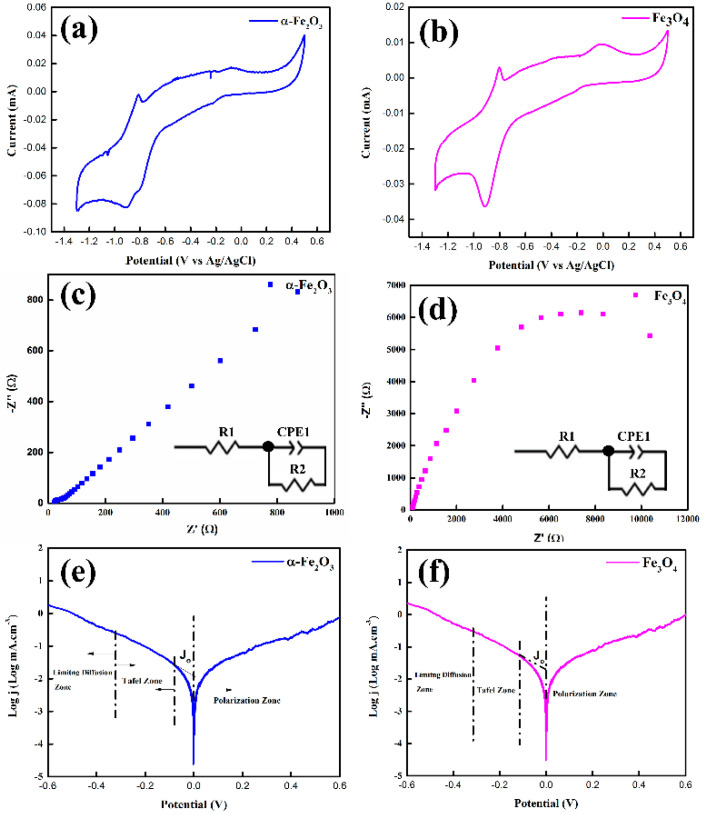
Cyclic voltammetry curves of (**a**) hematite (α-Fe_2_O_3_) and (**b**) magnetite (Fe_3_O_4_) nanoparticles, EIS spectra of (**c**) hematite (α-Fe_2_O_3_) and (**d**) magnetite (Fe_3_O_4_) nanoparticles; tafel polarization curve of (**e**) hematite (α-Fe_2_O_3_) and (**f**) magnetite (Fe_3_O_4_) nanoparticles.

**Figure 7 nanomaterials-12-01635-f007:**
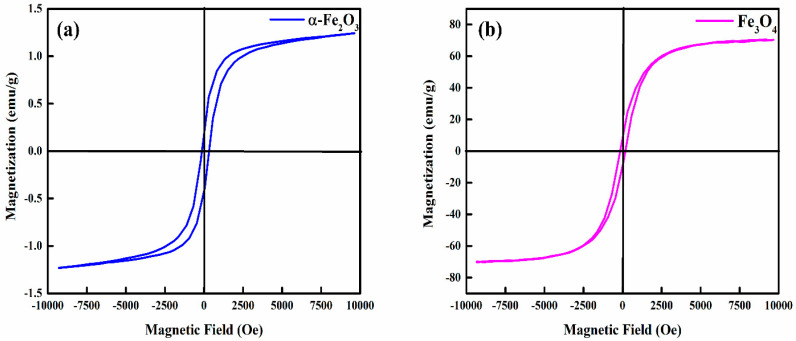
M-H curves for (**a**) hematite (α-Fe_2_O_3_) and (**b**) magnetite (Fe_3_O_4_) nanoparticles.

## Data Availability

The data presented in this study are available on request from the corresponding author.

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
