# Peer review of "Systematic Investigation of Structural, Morphological, Thermal, Optoelectronic, and Magnetic Properties of High-Purity Hematite/Magnetite Nanoparticles for Optoelectronics"

_nanomaterials, 2022, doi:10.3390/nano12101635_

Round 1

Author Response

Thank you very much for your careful review, useful comments, and kind recommendations on the structure of our manuscript. We have carefully revised the manuscript in light of your comments and a point by point response is given in the attached file.

Reviewer 2 Report

The α-Fe2O3 and Fe3O4 have attract intensive interesting because of their various applications. While it’s very difficult to synthesized high purity of these two materials with nanoscale. In this manuscript, the authors have synthesized high purity hematite and magnetite nanoparticles with a simple technique. They have investigated the structural, morphological, thermal, optoelectrical, and magnetic properties of these synthesized samples. They show that the nanomaterials with high values of coercivity. The results are interesting and the present observations are certainly of high quality. Here, there are some comments must be addressed before this interested paper publication.

  1. The introduction is too descriptive, which fails to guide the readers towards the key questions from previous works and thus realize the scientific significance of this work.
  2. The reviewer suggested the author divide the first two paragraph into one parts, point out the unsolved issues in the related research field, and highlight the scientific significance of this work.
  3. This is interesting paper, I suggest the author ask a native speaker polish their English and make the paper more readable and attractive.
  4. In this paper, the author provides the high quality SEM image, while with only very simplified description.  The morphology change can affect the physical properties of the nanomaterials, the author should briefly discussion, the relationship between the microstructure and the physical properties (Phys. Rev. Lett. (2022) 128, 015701; Science. 375 (2022) 1261).
  5. The author should discuss and highlight the new discovery and difference compare with others. This related discussion will make the paper more comprehensive and more attractive.
  6. By performing different physical properties measurement, the reviewer suggested the author provide more discussion, analysis. Provide in-depth relationship between the TGA Curve, UV-Vis Absorption Spectra, Electrochemical Measurements, not just list their experiments results.
  7. Overall, the logic and relations between each figure have not been clarified. A proper transition between different figures will help the readers to follow.
  8. The reviewer suggested the author re-organized the figures and make them more attractive.

Round 2

Reviewer 1 Report

The revision has led to an improvement of the manuscript. The introduced discussion section is rather application-related instead of providing a coherent physical picture, but this is up to the authors.

Reviewer 2 Report

After carefully read the revised manuscript many times, I am impressed with the effort that the authors have put into the revision of their manuscript. They have satisfactorily addressed my major concerns, thus, I believe this interesting work is worth of publication.